# Severity of Hepatocyte Damage and Prognosis in Cirrhotic Patients Correlate with Hepatocyte Magnesium Depletion

**DOI:** 10.3390/nu15112626

**Published:** 2023-06-03

**Authors:** Simona Parisse, Alessandra Gianoncelli, Gloria Isani, Francesco Luigi Gambaro, Giulia Andreani, Emil Malucelli, Giuliana Aquilanti, Ilaria Carlomagno, Raffaella Carletti, Monica Mischitelli, Flaminia Ferri, Veronica Paterna, Quirino Lai, Gianluca Mennini, Fabio Melandro, Cira Di Gioia, Massimo Rossi, Stefano Iotti, Michela Fratini, Stefano Ginanni Corradini

**Affiliations:** 1Department of Translational and Precision Medicine, “Sapienza” University of Rome, Viale dell’Università 37, 00185 Rome, Italy; simona.parisse@uniroma1.it (S.P.); monica.mischitelli@uniroma1.it (M.M.); flaminia.ferri@uniroma1.it (F.F.); paterna.1703481@studenti.uniroma1.it (V.P.); 2Elettra-Sincrotrone Trieste, Strada Statale 14 km 163,5 in AREA Science Park, Basovizza, 34149 Trieste, Italy; alessandra.gianoncelli@elettra.eu (A.G.); giuliana.aquilanti@elettra.eu (G.A.); ilaria.carlomagno@elettra.eu (I.C.); 3Department of Veterinary Medical Sciences, Alma Mater Studiorum-University of Bologna, Via Tolara di Sopra 50, 50055-Ozzano dell’Emilia, 40064 Bologna, Italy; gloria.isani@unibo.it (G.I.); giulia.andreani2@unibo.it (G.A.); 4Department of Radiological Sciences, Oncology and Pathological Anatomy, “Sapienza” University of Rome, Viale del Policlinico 155, 00161 Rome, Italy; francescoluigi.gambaro@uniroma1.it (F.L.G.); raffaella.carletti@uniroma1.it (R.C.); cira.digioia@uniroma1.it (C.D.G.); 5Department of Pharmacy and Biotechnology, University of Bologna, 40126 Bologna, Italy; emil.malucelli@unibo.it (E.M.); stefano.iotti@unibo.it (S.I.); 6General Surgery and Organ Transplantation Unit, “Sapienza” University of Rome, Viale del Policlinico 155, 00161 Rome, Italy; quirino.lai@uniroma1.it (Q.L.); gianluca.mennini@uniroma1.it (G.M.); massimo.rossi@uniroma1.it (M.R.); 7National Institute of Biostructures and Biosystems, Via delle Medaglie d’oro, 305, 00136 Rome, Italy; 8CNR-Institute of Nanotechnology c/o Physics Department, Sapienza University of Rome, Piazzale Aldo Moro 7, 00185 Rome, Italy; michela.fratini@gmail.com; 9Laboratory of Neurophysics and Neuroimaging (NaN), IRCCS Fondazione Santa Lucia, Via Ardeatina 306, 00179 Rome, Italy

**Keywords:** hepatocyte, liver, liver transplantation, MELD, MELDNa, transient receptor potential melastatin 7, TRPM7

## Abstract

We aimed to evaluate the magnesium content in human cirrhotic liver and its correlation with serum AST levels, expression of hepatocellular injury, and MELDNa prognostic score. In liver biopsies obtained at liver transplantation, we measured the magnesium content in liver tissue in 27 cirrhotic patients (CIRs) and 16 deceased donors with healthy liver (CTRLs) by atomic absorption spectrometry and within hepatocytes of 15 CIRs using synchrotron-based X-ray fluorescence microscopy. In 31 CIRs and 10 CTRLs, we evaluated the immunohistochemical expression in hepatocytes of the transient receptor potential melastatin 7 (TRPM7), a magnesium influx chanzyme also involved in inflammation. CIRs showed a lower hepatic magnesium content (117.2 (IQR 110.5–132.9) vs. 162.8 (IQR 155.9–169.8) μg/g; *p* < 0.001) and a higher percentage of TRPM7 positive hepatocytes (53.0 (IQR 36.8–62.0) vs. 20.7 (10.7–32.8)%; *p* < 0.001) than CTRLs. In CIRs, MELDNa and serum AST at transplant correlated: (a) inversely with the magnesium content both in liver tissue and hepatocytes; and (b) directly with the percentage of hepatocytes stained intensely for TRPM7. The latter also directly correlated with the worsening of MELDNa at transplant compared to waitlisting. Magnesium depletion and overexpression of its influx chanzyme TRPM7 in hepatocytes are associated with severity of hepatocyte injury and prognosis in cirrhosis. These data represent the pathophysiological basis for a possible beneficial effect of magnesium supplementation in cirrhotic patients.

## 1. Introduction

Magnesium (Mg) is the fourth most abundant cation in the human body (Ca > K > Na > Mg) and the second most abundant intracellular cation after potassium. About 99% of the total body Mg^2+^ is located in the intracellular milieu, while around 1% is present in blood and extracellular fluids [1,2]. 

The complex role of Mg in cell and tissue metabolism is multifactorial. Established evidence show that Mg acts mainly as a key signalling element in cell physiology; therefore, the concept about Mg as electrolyte is simplistic and outdated [3,4,5,6,7]. Mg is involved in most metabolic and biochemical pathways and is required in a wide range of vital functions, such as bone formation, neuromuscular activity, signalling pathways, bioenergetics, glucose, lipid and protein metabolism, DNA and RNA metabolism and stability, cell proliferation and differentiation [3,4,5,6,7,8,9]. The enzymatic databases list more than 600 enzymes with Mg indicated as a cofactor, and additional 200 are reported, in which Mg acts as an activator [10]; however, it should be specified that since Mg interacts directly with the enzyme substrates, it is itself a substrate rather than a cofactor [5,6,7].

Mg deficiency has been documented in several pathological conditions since the end of the last century [11]. Serum Mg concentration does not reflect the amount of magnesium in different body tissues. Therefore, a normal concentration of serum Mg does not rule out Mg deficiency [12]. In recent decades, a great number of epidemiological, clinical, and experimental research papers have documented that hypomagnesemia and/or chronic Mg deficiency may result in disturbances in nearly every organ/body, contributing to or exacerbating pathological consequences and causing potentially fatal complications [13,14]. 

It has been proposed that Mg depletion could be implicated in the pathogenesis and progression of chronic liver diseases [15,16]. This hypothesis is based on studies that have associated a low intake or depletion of Mg with liver disease severity and the administration of Mg-containing compounds with the improvement of hepatic inflammation and fibrosis. In fact, in three large human studies, low dietary Mg intake was associated with more severe hepatic fibrosis [17], increased mortality due to liver disease [18], and increased risk of non-alcoholic fatty liver disease [19]. Furthermore, in experimental studies, Mg depletion has been shown to: (a) induce an increase in the deposition of collagen in vivo in the liver of rats treated with CCL4 or with ethanol [20]; (b) induce lipid accumulation in hepatocytes, inflammation and ballooning in rat liver in vivo [21]; and (c) alter the respiratory and metabolic functions of hepatocyte mitochondria in vitro [22]. Finally, human studies have also shown a beneficial effect of the administration of Mg-containing compounds on serum aminotransferase activities of patients with alcoholic liver disease [23,24], obesity [25] or in the setting of drug-induced liver injury [26]. In agreement, other experimental studies have shown that the administration of Mg in different types of formulations in different models of liver damage has beneficial effects on hepatocyte death, hepatic inflammation [27,28,29,30] and fibrosis [31,32]. Furthermore, experimental studies have shown that an antifibrotic mechanism of several Mg formulations is the inhibition of hepatic stellate cells (HSC) activity [31,33,34]. 

To support the hypothesis that Mg depletion may be implicated in the pathogenesis and progression of chronic liver disease, it would also be necessary to demonstrate that tissue Mg content is low in these patients and progressively decreases with increasing the disease severity. As for the studies performed on humans, most of them [35,36,37,38,39,40], although not all [41,42], found lower blood concentrations of Mg in cirrhotic patients than in healthy controls. Similarly, most [38,40,43,44] studies, but not all [35,37,42], have found a progressive reduction in magnesemia that correlated with the worsening of cirrhosis. In a study using the Mg intravenous loading test, a body depletion of Mg was clearly demonstrated in advanced cirrhosis compared to healthy controls [36]. Accordingly, studies investigating the tissue content of Mg in cirrhotic patients found its depletion in sublingual epithelial cells [39], red blood cell, mononuclear and polymorphonuclear cells [35], and muscles [41,45]. As for the muscle depletion of Mg, it was also found in the presence of normal serum Mg concentrations [41] and correlated with the severity of the disease [45]. Only one study, conducted on pediatric patients with secondary cryptogenic or biliary cirrhosis [46], investigated the liver Mg content using atomic absorption spectrometry and found that it was reduced compared to subjects with healthy liver undergoing exploratory laparatomy, without correlation with serum Mg concentrations. Regarding adult patients with cirrhosis of more frequent etiology, no data are available on the Mg content of the liver tissue as a whole and, as recently pointed out by Liu et al. [16], within the hepatocytes. 

A major Mg cellular transport mechanism depends on the transient receptor potential melastatin-subfamily member 7 (TRPM7) chanzyme. TRPM7 is ubiquitously expressed; has an ion channel permeable to the divalent cations Mg^2+^, Ca^2+^ and Zn^2+^; and also possesses a C-terminal α-kinase domain able to phosphorylate substrates and to regulate stability and localization of its own channel [47,48]. TRPM7 is widely involved in the cell cycle and metabolism, inflammation and various cell death processes [47,49] and, as regards the liver, hepatocellular necrosis, lobular inflammation, increased activity of serum aminotransferases, activation of hepatic stellate cells and fibrosis [50]. For these reasons, a relevant role of TRPM7 in the pathogenesis of liver disease and cirrhosis has been hypothesized [50,51]. TRPM7 expression has been demonstrated in rat hepatocyte cell lines, human hepatoma cells and, in vivo, in human hepatocytes but has not been correlated with liver disease [52,53]; therefore, in the present study, we compared, in samples obtained at the time of liver transplantation, the Mg content in liver tissue and the hepatocyte expression of TRPM7 in healthy liver of donors and in hepatic cirrhotic tissue obtained from adult patients. In cirrhotic patients, we also correlated tissue and hepatocyte Mg content and hepatocyte TRPM7 expression with a widely used disease prognostic index, the MELDNa score, and with an index of hepatocyte damage, such as serum aminotransferase activity [54]. 

## 2. Materials and Methods

### 2.1. Study Design

This is a single-center retrospective study relating Mg content in liver and hepatocytes, and TRPM7 expression in hepatocytes to clinical data of patients with cirrhosis who underwent liver transplantation at the Sapienza University of Rome, Azienda Ospedaliero Universitaria Policlinico Umberto I, Rome, Italy. The study was performed in accordance with the Declaration of Helsinki and approved by the Ethics Committee of Sapienza University–Policlinico Umberto I (Ref. No. 1129/14.12.06). 

### 2.2. Study Population and Data Collection 

Samples of cirrhotic liver tissue taken at the time of liver transplantation from 58 patients were retrospectively analyzed. Exclusion criteria were acute on chronic liver failure, age > 70 years, magnesium supplementation, estimated glomerular filtration rate < 50, alcohol intake discontinued for <6 months, treatment with vitamin D, calcium or bisphosphonates, and thyroid disease. Twenty-six liver samples without steatosis, obtained from deceased donors, were analyzed as healthy liver controls. 

Wedge liver biopsies were performed on the left hepatic lobe immediately after liver explant of cirrhotic livers and before ischemia in donors. The cirrhotic study population was divided according to the availability of snap frozen samples or formalin-fixed paraffin sections. In 27 cirrhotic patients (study population A) and 16 deceased liver donors, from whom snap frozen liver biopsy was available, we measured the tissue Mg content by atomic absorption spectroscopy (Figure 1). In 31 cirrhotic patients (study population B) and 10 deceased liver donors, from whom paraffin-preserved liver biopsy was available, we measured the cytoplasmic expression of TRPM7 in hepatocytes by immunohistochemistry. Furthermore, in 15 of the cirrhotic patients of study population B, we measured the intrahepatocellular Mg content using the TwinMic beamline of Elettra Synchrotron in Trieste 65 and, in 7 of these, also at the X-ray Fluorescence (XRF) beamline 66 at the same synchrotron facility. 

Demographic and clinical data of cirrhotic patients were retrospectively collected from outpatient and hospitalization records at the time of transplantation. Data regarding deceased liver donors were collected from medical records at the time of transplantation. All measurements performed on cirrhotic livers were performed blinded to patient demographic and clinical variables. MAFLD-associated cirrhosis was defined by the presence of previous histological or imaging evidence of hepatic steatosis and at least one of the following characteristics: (a) type 2 diabetes; (b) overweight/obesity (dry weight BMI > 25 kg/m^2^); (c) evidence of metabolic abnormalities defined by the presence of at least two of the following criteria: 1. Fasting glucose levels 100–125 mg/dL or antidiabetic treatment; 2. Triglycerides > 150 mg/dL or specific drug treatment; 3. High-density lipoprotein cholesterol < 50 mg/d in women or <40 mg/dL in men, or specific drug treatment; 4. Blood pressure > 130/85 mm Hg or specific drug treatment. Since waist circumference, high sensitivity C reactive protein and homeostasis model assessment index were not available in many patients, they were not taken into account. Alcohol-associated cirrhosis was diagnosed in the presence of daily ethanol intake exceeding 30 g for men and 20 g for women, for more than 10 years.

### 2.3. Magnesium Quantification in Liver Tissue Using Atomic Absorption Spectrometry

To avoid contamination, polyethylene disposables were thoroughly washed with HCl 1 N under a fume hood, and disposable gloves were worn during the procedure. All the reagents were purchased from Merck (Merck-Sigma-Aldrich, Darmstadt, Germany); the acids were of Suprapur grade. Biopsies of liver tissue were placed in individual acid-washed teflon jars and were digested with 2 mL 65% HNO_3_ and 0.5 mL 30% H_2_O_2_ in a microwave oven for 5 min at 250 W, 5 min at 400 W, 5 min at 500 W and 1 min at 600 W. The cooled samples were transferred into polyethylene volumetric flasks, diluted to 10 mL and directly analyzed using a flame atomic spectrophotometer equipped with a deuterium lamp background correction (AAnalyst 100, Perkin Elmer, Waltham, MA, USA) for Mg determination. All the samples were run in batches, which included blanks; there was no evidence of any contamination in these blanks. The accuracy of the method was evaluated using the analysis of international standards (ERM^®^-BB422 fish muscle). The concentrations found with the method used in this study fell into the certified uncertainty interval given by ERM, corresponding to a 95% confidence level. The Mg detection limit for flame atomic spectrophotometry was 0.04 μg/mL. Mg content in biopsies of liver tissue were reported as μg/g wet weight (w.w.). 

### 2.4. Determination of Magnesium in Hepatocytes by Synchrotron Based X-ray Fluorescence Microscopy

Synchrotron-based X-ray fluorescence (XRF) microscopy to assess hepatocellular Mg content was performed using paraffin-embedded sections. Consecutive sections of the same specimen were used to obtain hematoxylin–eosin and 4 µm thick ultralene-mounted sections for XRF microscopy and low-energy micro-X-ray fluorescence (microXRF) supplementary measurements [55] analyses. MicroXRF measurements combined with scanning transmission X-ray microscopy were carried out at TwinMic beamline to retrieve information at the single-cell level [55,56]. The regions of interest were selected by inspecting the eosin and hematoxylin stained sections of the corresponding tissues. The incident X-ray beam energy was set to 1.5 keV to ensure the best excitation and detection of the Kα lines of Mg and Na atoms. The samples were raster-scanned with a step size of 1µm across a microprobe of 1 µm diameter delivered perpendicularly to the sample plane by a 600 µm diameter Au zone plate optics with 50 nm outermost zone. 8 Silicon Drift Detectors located at 20 degrees in respect to the sample plane collected the XRF photons emitted by the sample [57] while a fast readout CCD camera acquired the transmitted X-ray photons producing simultaneous absorption and differential phase contrast images of the analyzed areas [58]. All experiments were performed in a high vacuum (HV) condition (10^−6^ mbar). To further confirm the TwinMic beamline results on Mg content in single hepatocytes, we performed complementary XRF microscopy analyses for Mg content in hepatocytes, including clusters of surrounding hepatocytes from the same sample. To do this we used XRF beamline on larger sample areas with a spatial resolution in the range of 100 µm. At the XRF beamline the samples were loaded in an Ultra HV chamber [59]. The beam energy was selected with a multilayer mirror, and the X-ray intensity was monitored using a polycrystalline diamond plate developed by the Elettra electronics group. The beam size was set to 200 × 100 µm^2^ (horizontal × vertical) and the X-ray fluorescence emission was measured in 45/45 geometry using a Bruker XFlash 5030 detector placed at 15 mm from the samples. The samples were raster-scanned using a seven-axes manipulator across an incident beam of 2.2 keV, to enhance the X-ray fluorescence emission of Mg. All XRF spectra were then processed using PyMCA software through a least-squares fitting algorithm and SNIP background substruction routine [60]. This algorithm, applied before the peak intensity estimation, ensures that the noise is removed and that the remaining peaks are statistically relevant. Mg hepatocellular content were reported as X-ray fluorescence counts.

### 2.5. Immunohistochemistry Analysis for TRPM7 in Hepatocytes

Immunohistochemical analysis for TRPM7 liver tissue expression was performed using paraffin-embedded sections obtained in both study group B cirrhotic liver specimens and organ donors. Normal parotid gland tissue from a single sample was used as a positive control to ensure quality and consistency of staining and to be confident that hepatocytes from study samples without stained granules in the cytoplasm were indeed considered negative. Due to the diversity between salivary gland ducts and liver parenchyma, we decided not to use comparison with the intensity of staining for TRPM7 in salivary ducts for semiquantitative assessment of TRPM7 expression in hepatocytes. Semiquantitative analysis of hepatocellular TRPM7 was performed after identifying at low magnification (10×) the areas of maximum expression, minimum expression and intermediate expression of each sample as a first step. In each of these three areas, analyzed separately at higher magnification (40×), the percentage of hepatocytes showing a “dot like” stain in the cytoplasm was recorded, and we graded the staining as intense or weak. We defined intense staining as a coarse granular cytoplasmic dark brown stain and weak staining as a finely granular cytoplasmic light brown stain. Hepatocytes without stained granules in the cytoplasm were considered negative. The total number of stained hepatocytes was considered as the sum of the intensely stained and weakly stained ones. Finally, for each sample, the average among the three areas was calculated for the percentage of hepatocytes with intense, weak and total staining. Nuclear and membrane staining, whenever present, was always barely perceptible; thus, it was disregarded. All immunohistochemistry readings for TRPM7 and those regarding the presence or absence of mild inflammatory infiltrate in liver donor biopsies were performed by one author (F.L.G.).

### 2.6. Statistical Analysis

No missing data relative to the study variables were observed. Data are presented as medians and interquartile ranges (IQR) for continuous variables and as numbers and percentages for discrete variables. For continuous variables, the normality was assessed by the Shapiro–Wilk test. The differences between two groups were evaluated by Mann–Whitney U test or t-test, and the chi-squared test or Fisher’s exact test, as appropriate. Pearson correlation coefficient was calculated to evaluate the correlation between clinical patients’ data and Mg liver or hepatocellular content, or hepatocellular TRPM7 expression. Hepatic Mg content and TRPM7 expression were investigated in relation to serum aminotransferase activity and, for cirrhotic patients, MELDNa measured at the time of liver transplantation [54]. In cirrhotic patients, the MELDNa data were available not only at the time of liver transplant but also at the time of waitlisting, allowing the calculation of Δ-MELDNa. Δ-MELDNa was calculated with the following formula: Δ-MELDNa = ((MELDNa at transplant—MELDNa at listing)/time elapsed between listing and transplant expressed in days) × 100. In this way, positive Δ-MELDNa values mean that MELDNa, and therefore the patient’s prognosis, at the time of transplantation were worse than at the time of listing, while Δ-MELDNa negative values indicate a clinical improvement during the time spent in the waiting list. Since MELDNa and serum AST activity at transplant and Δ-MELDNa were correlated, in the two investigated study population groups of cirrhotic patients, risk factors for increased MELDNa and serum AST at transplant and Δ-MELDNa were explored using separate multivariable linear regression models. In all models, we included those significantly associated with the dependent variable and those in other studies that have been shown to be associated with serum Mg concentration as initial independent covariates. The variables to be used for constructing the models were preliminarily selected using a least absolute shrinkage and selection operator (LASSO) regression (stepwise regression with backward elimination), with the intent to create parsimonious models in terms of number of covariates [61]. After having identified the variables to be analyzed in the models by backward elimination, a 1000-fold bootstrap (resampling with replacement) method was used. Beta coefficients and 95.0% confidence intervals (95.0% CI) were reported. A *p*-value < 0.05 was considered statistically significant. Statistical analyses were conducted using SPSS 27.0 (SPSS Inc., Chicago, IL, USA).

## 3. Results

### 3.1. In Liver Cirrhosis, Hepatic Magnesium Content Is Lower, and TRPM7 Expression in Hepatocytes Is Higher Than in Healthy Liver

Table 1 shows the demographic and clinical characteristics of cirrhotic patients in study populations A and B and their respective healthy liver donor control groups. 

The only significant difference between the two populations of cirrhotic patients was the lower frequency of viral etiology of the disease in study population B compared to study population A, while no difference was present for age, sex, BMI, metabolic-Metabolic-associated fatty liver disease (MAFLD) and alcoholic etiology of cirrhosis, HCC, diuretic treatment, serum aminotransferase activities, MELDNa both at listing and at transplant, length of time on the waiting list, and Δ-MELDNa. The two populations of cirrhotic patients did not differ from the respective healthy liver donor control groups for age, sex, BMI and serum aminotransferase activities. The median content of Mg, measured in biopsies of liver tissue using atomic absorption spectrometry, was significantly (*p* < 0.001) lower in cirrhotic patients of the study population A (117.2 (IQR 110.5–132.9) μg/g w.w.) compared to its healthy liver donor controls group (162.8 (IQR 155.9–169.8) μg/g w.w.) (Figure 2). 

Regarding TRPM7 expression, Figure 3A is representative of the staining in human salivary glands with highly positive cells with large cytoplasmic granules that were used as a positive control. Figure 3B,C, is representative of TRPM7 staining in hepatocytes of cirrhotic livers from study population B with some cells stained intensely, some weakly, and others unstained. Then, we applied a semiquantitative scoring system by calculating the percentage of hepatocytes with intense or weak staining in cirrhotic and control donor livers. In 25 of 31 (81%) cirrhotic liver biopsies from study population B there was intense staining in some hepatocytes, with the maximum percentage of hepatocytes affected by heavy staining of 31% in one patient. Weak staining affected all cirrhotic liver samples, and 45% of these had at least half of the hepatocytes affected. In contrast, only 3 of 10 (30%) of the donor control liver biopsies showed intense staining in any hepatocyte, with the maximum percentage of hepatocytes affected by intense staining of 3% in one donor. Weak staining affected all donor liver samples, but only 10% of these had at least half of the hepatocytes affected.

The cirrhotic group of study population B, compared to the its healthy liver donor control group, showed significantly higher percentages of hepatocytes with intense TRPM7 staining (2.8 (0.2–9.2) vs. 0.0 (0.0–1.8)%; *p* < 0.01), with weak staining (47.4 (34.0–56.2) vs. 18.8 (10.7–32.8)%; *p* < 0.01), and with total staining, i.e., weak or intense, (53.0 (36.8–62.0) vs. 20.7 (10.7–32.8)%; *p* < 0.001) (Figure 4A–C). 

At least 4% of hepatocytes with intense TRPM7 staining were present in 45% of the cirrhotic liver samples and in none of the donor samples. Overall, 6 of 10 donor livers had a mild mononuclear cell portal inflammatory infiltrate. Interestingly, in donor livers the percentage of TRPM7 positive hepatocytes was significantly lower (*p* = 0.01) in those without any histological sign of inflammation (10.3 (IQR 8.4–13.3)%; *n* = 4) versus those with mild inflammation (30.0 (IQR 21.2–43.6)%; *n* = 6).

### 3.2. In Liver Cirrhosis, MELDNa at the Time of Liver Transplantation Is Inversely Correlated with the Content of Magnesium in Liver Tissue and Hepatocytes and Directly with the Hepatocellular Expression of TRPM7

In cirrhotic patients of study population A, the MELDNa score measured at the time of liver transplant correlated inversely with the content of Mg measured in liver biopsies using atomic absorption spectrometry (Figure 5A). To verify if this inverse correlation was retained, we used the X-ray fluorescence technique, an analytical method that allowed us to measure the Mg inside the hepatocytes in 15 cirrhotic patients of study population B. The significant inverse correlation between MELDNa and hepatocyte content of Mg measured at TwinMic beamline (r = −0.531; *p* = 0.042; Figure 5B) and, in 7 of these patients, measured at XRF beamline (r = −0.854; *p* = 0.014) (Figure 5C) was confirmed. Furthermore, the MELDNa score of study population B correlated directly with the percentage of hepatocytes with the intense and total, i.e., sum of intense and weak, staining for TRPM7 (Figure 5D,E). No correlation was found between MELDNa and the percentage of hepatocytes weakly stained for TRPM7.

We then wanted to verify whether the correlations of the MELDNa values with Mg content in liver biopsies and with the hepatocellular expression of TRPM7 were independent of confounders. To do this, we first analyzed whether patients differed within each of the two cirrhotic study populations when subgrouped into low and high MELDNa, based on their respective median MELDNa value (Appendix A). 

As expected, in the study population A, patients with high MELDNa had significantly (*p* = 0.005) lower Mg content than the low MELDNa group (113.2 (IQR 101.7–123.2) μg/g vs. 125.8 (IQR 116.8–141.1) μg/g). In study population B, patients with high MELDNa showed a significantly higher proportion of hepatocytes with intense (9.0 (IQR 3.8–17.3)% vs. 0.8 (IQR 0.0–1.8)%; *p* < 0.001) and total (55.9 (IQR 51.5–75.7)% vs. 47.2 (IQR 24.8–55.0)%; *p* = 0.014) TRPM7 staining than those with low MELDNa. No intergroup difference in study population B was found with regard to the percentage of hepatocytes with weak TRPM7 expression. Furthermore, in both study populations A and B (Appendix A), patients with lower MELDNa had a significantly higher frequency of HCC and a lower frequency of diuretic treatment compared to patients with higher MELD. No intergroup differences were present for age, sex, BMI and etiology of cirrhosis. The serum Mg concentration, available only for study population B, did not differ in patients with low compared to high MELDNa. The inverse correlation between MELDNa and the Mg content measured in liver biopsies and the direct correlations between MELDNa and the percentage of hepatocytes with intense or total TRPM7 staining were significant also performing multiple linear regression (Table 2).

Finally, in the cirrhotic study population B, the serum Mg concentration correlated directly (r = 0.553; *p* = 0.033) with the Mg content in hepatocytes measured at the TwinMic beamline (Appendix A), but did not correlate with any degree of TRPM7 expression. Furthermore, TRPM7 expression did not correlate with hepatic Mg content, measured at both Elettra Synchroton TwinMic and XRF beamlines.

### 3.3. In Liver Cirrhosis, Serum AST Activity at the Time of Liver Transplantation Is Inversely Correlated with Magnesium Content in Liver Tissue and Hepatocytes and Directly with Hepatocellular Expression of TRPM7

In cirrhotic patients of the group “study population A”, the serum activity of AST at transplant, but not that of ALT, was inversely correlated with the Mg content measured in liver biopsies using atomic absorption spectrometry (r = −0.458; *p* = 0.016; Figure 6A). In cirrhotic patients of study population B, the serum activity of AST at transplant, but not that of ALT, was inversely correlated with the content of Mg in hepatocytes measured using both the TwinMic (r = −0.615; *p* = 0.015; Figure 6B) and XRF beamlines (r = −0.758; *p* = 0.048; Figure 6C). In the cirrhotic patients of the group “study population B”, serum activity of AST directly correlated with the percentage of both high intensity (r = 0.511; *p* = 0.003; Figure 5D) and total (r = 0.428; *p* = 0.016; Figure 5E) but not with the percentage of weakly TRPM7 positive hepatocytes. Serum activity of ALT directly correlated with the percentage of both high intensity (r = 0.439; *p* = 0.013) and total (r = 0.396; *p* = 0.028) but not with the percentage of weakly TRPM7-positive hepatocytes.

We then wanted to verify whether the correlations of the serum AST activity with Mg content in liver biopsies and with the hepatocellular expression of TRPM7 were independent of confounders. To do so we first analyzed whether patients differed within each of the two cirrhotic study populations in clinical and demographic characteristics when grouped according to the respective AST median value (Appendix A). In the group “study population A”, the content of Mg measured in liver biopsies using atomic absorption spectrometry did not differ between the group with low and high AST serum activity. In the group “study population B”, compared to patients with lower values, those having high AST activity showed a significantly higher percentage of hepatocytes with intense, but not weak or total, TRPM7 staining. No intergroup differences were present for sex, BMI, etiology of cirrhosis and diuretic use in both study populations. The serum Mg concentration, available only for study population B, did not differ in patients with low compared to those with high serum AST. The inverse correlation between serum AST activity and the Mg content measured in liver biopsies and the direct correlation between AST serum activity and the percentage of hepatocytes with intense, but not total, TRPM7 staining remained significant even at multiple linear regression (Table 3).

### 3.4. In Liver Cirrhosis, the Worsening of MELDNa during the Waitlist Time Is Inversely Correlated with the Content of Magnesium in Biopsies of Liver Tissue and Directly with the Hepatocellular Expression of TRPM7 at the Time of Liver Transplantation

In cirrhotic patients, the MELDNa data were available not only at the time of liver transplant but also at the time of waitlisting, allowing the calculation of Δ-MELDNa. Δ-MELDNa correlated negatively with the content of Mg in liver biopsies measured using atomic absorption spectrometry in the group “study population A” (r = −0.404; *p* = 0.037), while no significant correlation was found in study population B measuring Mg liver content at both TwinMic (r = −0.367; *p* = 0.178) and XRF beamlines (r = −0.652; *p* = 0.112). In the cirrhotic patients of the group “study population B”, Δ-MELDNa directly correlated with the percentage of both high intensity (r = 0.569; *p* = 0.001) and total (r = 0.402; *p* = 0.025), but not with the percentage of weakly stained, TRPM7-positive hepatocytes assessed at liver transplant. 

We then wanted to verify whether the correlations of Δ-MELDNa with Mg content in liver biopsies measured using atomic absorption spectrometry and with the expression of TRPM7 were independent of confounders. To do so we first analyzed whether patients differed within each of the two cirrhotic study populations in clinical and demographic characteristics when grouped according to the respective Δ-MELDNa median value (Appendix A). In the group “study population A”, the content of Mg measured in liver biopsies did not differ in the groups with low or high Δ-MELDNa. In study population B, patients with high compared to those with low Δ-MELDNa showed a significantly higher percentage of hepatocytes with intense and total, but not weak, TRPM7 staining. No intergroup differences were present for sex, BMI, etiology of cirrhosis and diuretic use in both study populations. The serum Mg concentration, available only for study population B, did not differ in patients with low compared to those with high Δ-MELDNa. At multiple linear regression, while introducing age, presence of HCC, serum Mg concentration, diuretic treatment as covariates in the initial backward elimination model, the direct correlation between Δ-MELDNa and the percentage of hepatocytes with intense (B = 0.436; 95% CI of B = 0.179–0.670; *p* = 0.009) and with total TRPM7 staining (B = 0.153; 95% CI of B = 0.054–0.270; *p* = 0.025) remained significant. On the contrary the Mg liver content was not significantly associated with Δ-MELDNa at multiple linear regression.

## 4. Discussion

In the present study, the median content of Mg measured in liver biopsies using atomic absorption spectroscopy at the time of liver transplantation is about 1/3 lower in cirrhotic patients than in the livers of healthy donor. This finding, obtained for the first time in adult patients with liver cirrhosis of common etiologies, is in agreement with a study performed in pediatric patients with secondary biliary cirrhosis or cryptogenic cirrhosis [46]. A second important difference found in cirrhotic patients is that the median of the total percentage of hepatocytes expressing TRPM7 in the cytoplasm is 2.6 times higher than in healthy donors. Furthermore, nearly half of the cirrhotic liver samples, but none of the donor samples, had at least 4% of hepatocytes heavily stained for TRPM7. 

Concerning cirrhotic patients, significant correlations were found between Mg content and TRPM7 expression in the liver, on the one hand, and simultaneously measured clinical variables, on the other. In particular, the MELDNa and the serum AST activity of patients at the time of liver transplantation correlated inversely with the content of Mg measured both in the liver biopsies using atomic absorption and within the hepatocytes analyzed by synchrotron XRF Microscopy. In addition, serum AST activity and MELDNa score measured at transplant and worsening of MELDNa while on the waiting list were directly correlated with the total percentage of hepatocytes expressing TRPM7 and with a percentage of them with intense expression. Correlations between Mg content in liver biopsies with MELDNa and serum AST measured at transplantation were independent of some confounding factors known to affect blood or intracellular Mg levels. Similarly, the correlation of TRPM7 intense staining with serum AST and the correlation of both TRPM7 intense staining and total TRPM7 staining with MELDNa at transplant and with the worsening of MELDNa while on the waiting list were also independent of confounders. These factors, included in the linear regression models as independent variables, were age [62], diuretic therapy [63], presence of HCC [64] and blood concentration of Mg, available only for models concerning the expression of TRPM7. The number of liver samples measured by synchrotron methods for the content of Mg in hepatocytes did not allow us to perform multivariate analyses.

Taken together, our results demonstrate a close link between hepatocyte Mg metabolism and the presence and severity of liver cirrhosis. Regarding the progression of cirrhosis, we found close links between hepatocyte Mg metabolism and two simultaneously measured clinical indices, such as predominantly mitochondrial hepatocyte damage measured by serum AST and short-term prognosis of cirrhosis assessed by MELDNa. In particular, as the hepatic inflammation and the severity of the prognosis increased, the Mg content in hepatocytes decreased and the expression of TRPM7, which facilitates the entry of Mg into these cells, increased. Strikingly then, the percentage of hepatocytes that, at the time of transplantation, strongly expressed TRPM7 was also associated with the recent time-dependent worsening of MELDNa while on the waiting list. 

Our hypothesis to explain these data is that, in cirrhosis, a hepatocellular Mg depletion occurs mainly caused by inflammation and that, in turn, the latter is increased by Mg depletion in a vicious circle similar to that demonstrated for stressful situations [65]. This hypothesis is strongly supported by data, both in animal and human models, demonstrating that stress induces Mg depletion in erythrocytes. Although these studies were performed with non-inflammatory stress stimuli, inflammation definitely constitutes stress and therefore inflammation is likely to induce Mg depletion [65]. Indeed, acute inflammation has also been shown to cause cellular Mg depletion. This has been demonstrated in a mouse model of muscle damage and, with respect to the liver, in both mice and humans secondary to acetaminophen-induced liver damage [66,67,68]. According to our hypothesis, cellular Mg depletion caused by inflammation, in turn, aggravates the latter. Accordingly, many experimental studies have demonstrated that Mg depletion induces inflammation in different organs, including the liver [21,67]. Among the multiple mechanisms by which Mg depletion can induce inflammation, the activation of TRPM7 can play an important role. Our data are in agreement with the previous demonstration of the presence of TRPM7 in human hepatocytes in vivo [53]. Hepatocyte Mg depletion causes TRPM7 overexpression to facilitate Mg entry into depleted cells [49]; however, TRPM7 activation is also able to stimulate inflammation and cell death, as demonstrated at several levels, including the liver cells [47,49,50,51]. It is noteworthy that in our study, we found a slight expression of TRPM7 in hepatocytes from donor livers, in which mild inflammation was present. In our study we did not find any correlation between TRPM7 expression and Mg content in hepatocytes. We hypothesize that this could be explained by the fact that there are many Mg transporters and, in the individual patient, the contribution of each of them to try to restore cellular Mg content may be different [15]. Previously, activation of TRPM7 only in hepatic stellate cells was thought to be responsible for liver fibrosis and elevation of serum aminotransferases in a mouse model of carbon tetracloride-induced fibrosis [51]. Although we did not focus on hepatic stellate cells in the present study, we found a close association of clinical indices with TRPM7 activation in hepatocytes. Our results suggest that the administration of Mg in cirrhotic patients could interrupt the vicious circle between Mg depletion in the liver and hepatic inflammation and that it could reduce the expression of TRPM7 in these cells with beneficial effects. This is consistent with the beneficial effect of administration of Mg-containing compounds demonstrated by other studies in alcoholics, in obese patients, in patients with drug-induced liver injury, and in experimental models of liver damage [23,24,25,27,30,31,32,33,34]. 

A limitation of our study concerns its retrospective design. Furthermore, we analyzed two different populations of patients with liver cirrhosis due to the availability of liver tissue and its storage mode. This was different for tissue Mg content measurement using atomic absorption spectrometry, on the one hand, and for hepatocyte Mg content measured with XRFM and TRPM7 expression, on the other hand; however, the fact that we found consistent results using two different analytical protocols to measure Mg in the liver, and at two different length scale, albeit in two different patient populations, strengthens our conclusions. Finally, we did not provide direct evidence that the cause of TRPM7 overexpression in the hepatocytes of cirrhotic patients is intracellular Mg depletion. In conclusion, in the present study, we demonstrated that, in human liver affected by cirrhosis compared to normal liver, hepatocytes are Mg depleted and show an overexpression of TRPM7, the channel for magnesium entry, which is probably secondary to Mg depletion. Furthermore, the degree of Mg depletion correlates with serum levels of AST, a marker of hepatocyte death, and with MELDNa, a prognostic severity score. The degree of hepatocyte overexpression of TRPM7 in hepatocytes also correlates not only with serum AST levels and MELDNa but also with the worsening of MELDNa over time. Finally, even in normal livers, TRPM7 is expressed in the presence of mild inflammation. These data suggest that Mg depletion and overexpression of TRPM7, which is known to have proinflammatory activity, may represent therapeutic targets in cirrhotic patients.

## Figures and Tables

**Figure 1 nutrients-15-02626-f001:**
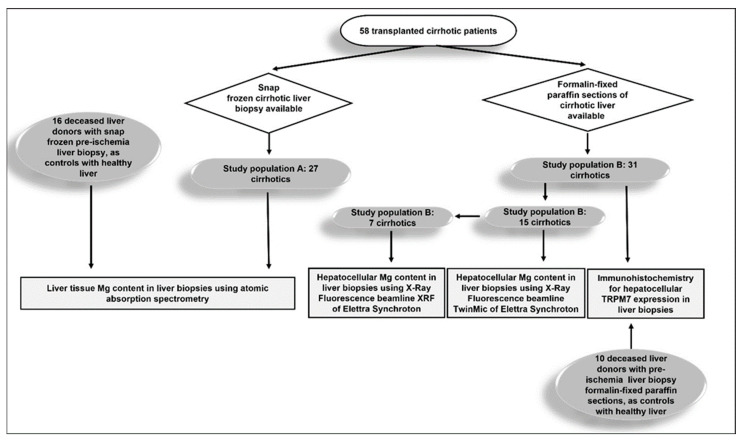
Schematic diagram depicting study populations of cirrhotic patients A and B and their respective healthy liver controls.

**Figure 2 nutrients-15-02626-f002:**
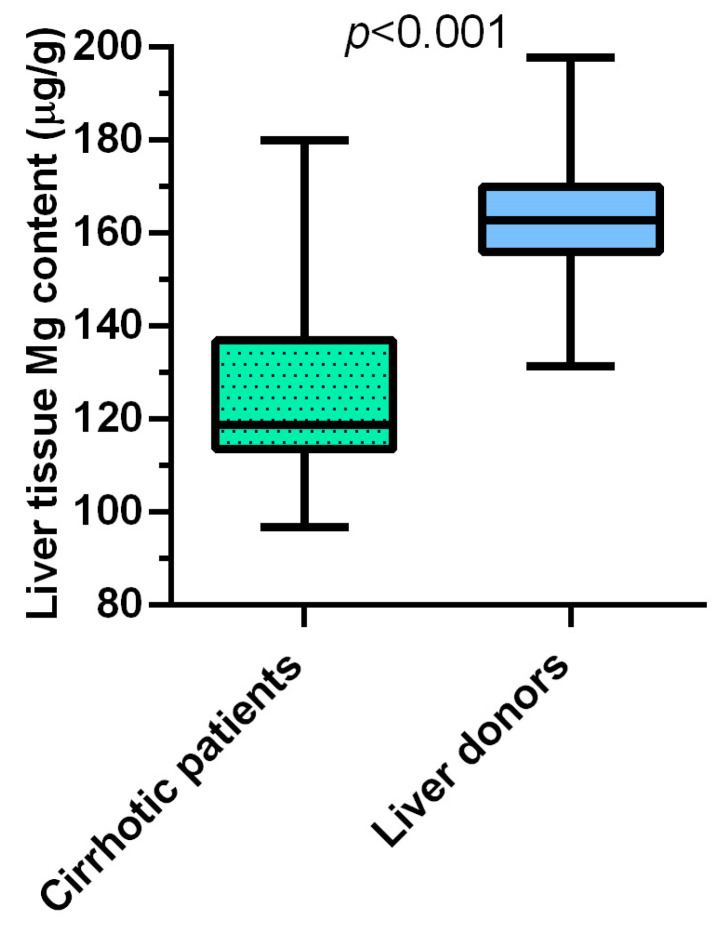
Box plots of the magnesium (Mg) content measured in liver biopsies using atomic absorption spectrometry in cirrhotic patients of study population A (*n* = 27) and in the respective healthy liver donor control group (*n* = 16).

**Figure 3 nutrients-15-02626-f003:**
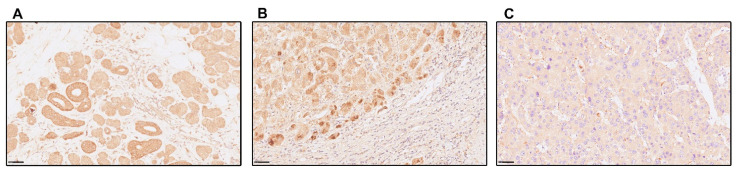
Representative immunohistochemistry images showing different expression patterns of transient receptor potential melastatin 7 (TRPM7) (40×; scale bars indicate 50 µm). Panel (**A**): human salivary glands with positive cells with large cytoplasmic granules are shown as a positive control; panel (**B**): cirrhotic liver of study population B showing hepatocytes with intense granular cytoplasmic staining and others with weak staining; Panel (**C**): cirrhotic liver of study population B showing few weakly stained and most unstained hepatocytes.

**Figure 4 nutrients-15-02626-f004:**
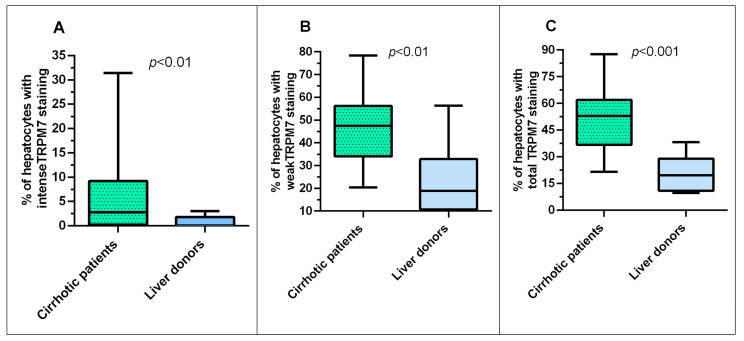
Box plots of the percentage of hepatocytes with TRPM7 expression in cirrhotic patients of study population B (*n* = 31) and in the respective healthy liver donor control group (*n* = 10) (panel (**A**): intense expression; panel (**B**): weak expression; panel (**C**): total expression, i.e., intense and weak).

**Figure 5 nutrients-15-02626-f005:**
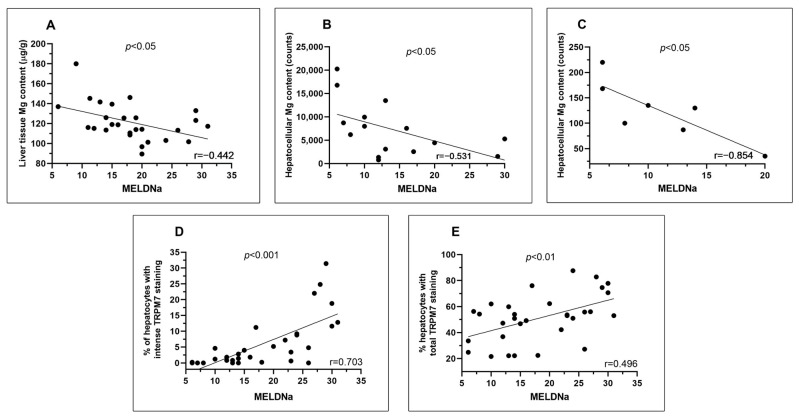
Correlation plots between MELDNa score of cirrhotic patients at liver transplant and the concentration of Mg measured in liver biopsies using atomic absorption spectrometry in study population A (panel (**A**); *n* = 27). Correlation plots between MELDNa score of cirrhotic patients at liver transplant and the hepatocyte content of Mg measured at Twinmic (panel (**B**); *n* = 15) or at XRF (panel (**C**); *n* = 7) beamlines of Elettra Synchrotron and the percentage of hepatocytes with intense (panel (**D**); *n* = 31) or total (panel (**E**); *n* = 31) TRPM7 staining in the study population B.

**Figure 6 nutrients-15-02626-f006:**
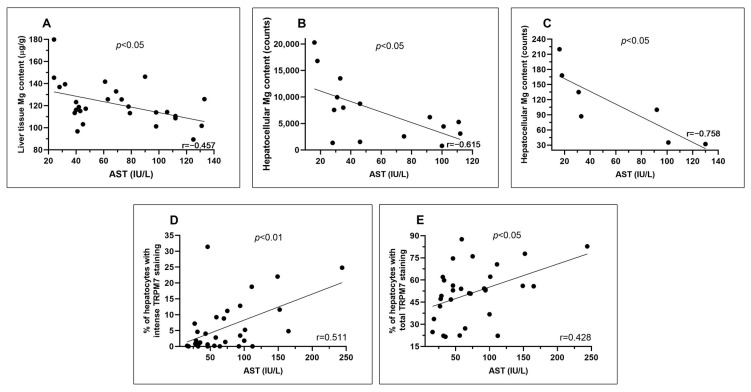
Correlation plots between serum AST activity of cirrhotic patients at liver transplant and the Mg content measured in liver biopsies using atomic absorption spectrometry in study population A (panel (**A**); *n* = 27). Correlation plots between serum AST activity of cirrhotic patients at liver transplant and their hepatocyte content of Mg measured at the TwinMic (panel (**B**); *n* = 15) or the XRF (panel (**C**); *n* = 7) beamlines of Elettra Synchrotron, and the percentage of hepatocytes with intense (panel (**D**); *n* = 31) or total (panel (**E**); *n* = 31) TRPM7 staining in the study population B.

**Table 1 nutrients-15-02626-t001:** Demographic and clinical characteristics of cirrhotic patients in study populations A and B and their respective healthy liver donor control group.

	Liver Donors (Controls of Study Population A)	Cirrhotic Patients of Study Population A	Liver Donors (Controls of Study Population B)	Cirrhotic Patients of Study Population B	*p* Value Liver Donors (Controls of Study Population A) vs. Cirrhotic Patients Study Population A	*p* Value Liver Donors (Controls of Study Population B) vs. Cirrhotic Patients Study Population B	*p* Value Cirrhotic Patients Study Population A vs. Cirrhotic Patients Study Population B
	16	27	10	31			
Age, median (IQR)	53 (35–61.5)	52.0 (48.0–60.0)	62,5 (36.0–75.8)	57.3 (52.3–63.2)	0.890	0.984	0.050
Sex, male, *n* (%)	11 (68.8)	22 (81.5)	6 (60.0)	27 (87.1)	0.460	0.082	0.720
BMI (Kg/m^2^), median (IQR)	25.6 (24.8–27.3)	24.3 (22.6–27.4)	25.5 (24.2–27.6)	25.9 (24.4–29.0)	0.880	0.869	0.210
Cirrhosis aetiology, *n* (%):	NA		NA		-	-	
MAFLD		16 (59.3)		17 (54.8)			0.735
Alcohol		10 (37)		16 (51.6)			0.266
Viral		19 (70.4)		12 (38.7)			0.016
HCC, yes (%)	0 (0)	13 (48.1)	0 (0)	17 (54.8)	<0.001	<0.001	0.820
Diuretic treatment, *n* (%):	NAV		NAV		-	-	0.938
None		8 (29.6)		11 (35.5)			
K sparing diuretics only		3 (11.1)		4 (12.9)			
Loop diuretics plus K sparing diuretics		16 (59.3)		16 (51.6)			
Serum AST (IU/L), median (IQR)	35.5 (26.3–47.3)	63.0 (40.0–98.0)	52.5 (35.0–237.5)	59.0 (33.0–100.0)	0.873	0.665	0.925
Serum ALT (IU/L), median, (IQR)	23 (18–30.5)	47.0 (30.0–60.0)	67.0 (20.0–102.3)	43.0 (25.0–70.0)	0.792	0.665	0.911
MELDNa, at the time of liver transplant, median (IQR)	NA	18.0 (14.0–21.0)	NA	17.0 (12.0–26.0)	NA	NA	0.938
MELDNa, at the time of listing, median (IQR)	NA	15.7 (11.5–18.1)	NA	16.5 (12.0–23.4)	NA	NA	0.803
Days on the waiting list before liver transplant, median (IQR)	NA	128.0 (49.0–390.0)	NA	125.0 (50.0–186.0)	NA	NA	0.294
Δ-MELDNa, median (IQR)	NA	0.18 (0.00–1.28)	NA	0.92 (−0.31–3.43)	NA	NA	0.692

Abbreviations: AST, aspartate amino transferase; ALT, alanine amino transferase; BMI, body mass index; HCC, hepatocellular carcinoma; IQR, interquartile range; MAFLD, metabolic associated fatty liver disease; MELDNa, model for end stage liver disease sodium; K, potassium; Mg, magnesium; NA, not applicable; NAV, not available.

**Table 2 nutrients-15-02626-t002:** Multivariate linear regression analyses of variables associated with MELDNa in cirrhotic patients in the two study populations A and B.

		Indipendent Variable	B	95% C.I. for B	*p* Value
**Study population A**	**Model 1 (Mg liver content measured by atomic absorption) ***	Liver Mg content (ug/g)	−0.102	−0.207–0.005	0.048
		HCC, yes	−6.757	−10.785–−2.571	0.008
**Study population B**	**Model 2 (percentage of hepatocytes with intense TRPM7 expression) ****	Percentage of hepatocytes with intense TRPM7 expression	0.297	0.175–0.578	0.005
		HCC, yes	−10.921	−13.683–−8.046	0.001
	**Model 3 (percentage of hepatocytes with total, i.e., weak and intense, TRPM7 expression) *****	Percentage of hepatocytes with total TRPM7 expression	0.101	0.023–0.179	0.021
		HCC, yes	−12.413	−15.274–−9.923	0.001

Regression analyses data are shown only for significant associations after backward elimination and bootstrapping of the best model. * Model 1: Age, HCC, diuretic treatment and Mg liver content measured by atomic absorption were introduced as covariates in the initial backward elimination model. ** Model 2: Age, presence of HCC, serum Mg concentration, diuretic treatment and the percentage of hepatocytes with intense TRPM7 expression were introduced as covariates in the initial backward elimination model. *** Model 3: Age, presence of HCC, serum Mg concentration, diuretic treatment and the percentage of hepatocytes with total TRPM7 expression were introduced as covariates in the initial backward elimination model.

**Table 3 nutrients-15-02626-t003:** Multivariate linear regression analyses of variables associated with serum AST activity in cirrhotic patients in the two study populations.

		Independent Variable	B	95% C.I. for B	*p* Value
**Study population A**	**Model 1** **(Mg liver content measured by atomic absorption) ***	Liver Mg content (ug/g)	−0.824	−1.560–−0.288	**0.007**
		Age, years	−0.946	−2.209–0.763	0.155
**Study population B**	**Model 2 (percentage of hepatocytes with intense TRPM7 expression) ****	Percentage of hepatocytes with intense TRPM7 expression	3.256	0.758–6.208	**0.040**
		Age, years	−2.787	−5.162–−0.501	**0.029**
	**Model 3 (percentage of hepatocytes with total, i.e., weak and intense, TRPM7 expression) *****	Percentage of hepatocytes with total TRPM7 expression	0.884	−0.078–1.958	0.114
		HCC, yes	−33.783	−65.886–−8.639	**0.041**

Regression analyses data are shown only for significant associations after backward elimination and bootstrapping of the best model. * Model 1: Age, HCC, diuretic treatment, and Mg liver content measured by atomic absorption were introduced as covariates in the initial backward elimination model. ** Model 2: Age, presence of HCC, serum Mg concentration, diuretic treatment and the percentage of hepatocytes with intense TRPM7 expression were introduced as covariates in the initial backward elimination model. *** Model 3: Age, presence of HCC, serum Mg concentration, diuretic treatment and the percentage of hepatocytes with total TRPM7 expression were introduced as covariates in the initial backward elimination model.

## Data Availability

Data are available upon request.

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
