# Peer review of "Severity of Hepatocyte Damage and Prognosis in Cirrhotic Patients Correlate with Hepatocyte Magnesium Depletion"

_nutrients, 2023, doi:10.3390/nu15112626_

Round 1
Reviewer 1 Report
Parisse et al et set out the experiment to evaluate the magnesium content in human cirrhotic liver and its correlation with serum AST levels, expression of TRPM7, and MELD-Na prognostic score. Report about the direct measurement of Mg content in the adult cirrhotic liver was scarce in previously literatures, their attempt was worthwhile to be performed. Nevertheless, there were many shortcomings in Methods/the way of data presentation, and some in Discussion.
Major concerns;
1. In Table 1: The definition of MAFLD and alcohol-associated cirrhosis should be included in Methods.
2. In Page 5: What was the difference in the indication of device to measure hepatocellular Mg, between the TwinMic beamline of Elettra Synchrotron in Trieste 65 and the X-ray Fluorescence (XRF) beamline 66?
3. In Page 5: Determination of Magnesium in hepatocytes by Synchrotron Based X-ray Fluorescence Microscopy: There were no documentations how to select the region of interest and to the way of statistical determination of counts.
4. In Page 5, Methods in Immunohistochemistry analysis for TRPM7 in hepatocytes: Even human salivary gland cells were used as positive control, how authors define intense/average/weak expression in hepatocytes? Moreover, as in Figure 4, why did authors regard total staining as intense or weak, not as intense and average?
5. In Page 9, lane 3-6, regarding TRPM7 staining: Did 81% mean 81% of patients? Did up to 31% of hepatocytes indicate that 31% out of total hepatocytes in a certain patient were TRPM7 intense positive? Again, such vague expression or shortcomings are likely due to the fact that Methods in Immunohistochemistry analysis for TRPM7 in hepatocytes in Page 5 did not define parameters of immunostaining rigorously.
6. In Page 10, lane 3-5, Who and how many pathologists diagnosed histology of donor livers (histological sign of inflammation)?
7. In Discussion, Page 18, lane 535-536: Were Mg content in hepatocytes negatively associated with TRPM7 staining in corresponding tissue sections?
Minor editing of English language required, but no significant issues detected.
Author Response
- In Table 1: The definition of MAFLD and alcohol-associated cirrhosis should be included in Methods.
MAFLD-associated cirrhosis was defined by the presence of previous histological or imaging evidence of hepatic steatosis and at least one of the following characteristics: a) type 2 diabetes; b) overweight/obesity (dry weight BMI >25 kg/m2); c) evidence of metabolic abnormalities defined by the presence of at least two of the following criteria: 1. Fasting glucose levels 100–125 mg/dL or antidiabetic treatment; 2. Triglycerides >150 mg/dL or specific drug treatment; 3. High-density lipoprotein cholesterol <50 mg/d in women or <40 mg/dL in men or specific drug treatment; 4. Blood pressure >130/85 mm Hg or specific drug treatment. Since waist circumference, high sensitivity C reactive protein and Homeostasis Model Assessment index were not available in many patients, they were not taken into account. Alcohol-associated cirrhosis was diagnosed in the presence of daily ethanol intake exceeding 30 g for men and 20 g for women, for more than 10 years. We have now added these definitions to the method section (lines 154-165 of the revised version).
- In Page 5: What was the difference in the indication of device to measure hepatocellular Mg, between the TwinMic beamline of Elettra Synchrotron in Trieste 65 and the X-ray Fluorescence (XRF) beamline 66?
The main difference between the two beamlines (TwinMic & XRF) is the spatial resolution. In this study, TwinMic provided a beam of 1 micron in diameter. Such beam is ideal for inspecting fine details of the element distribution at cellular level of a single cell. On the other hand, the XRF beamline delivered a larger (200 x 100 µm2) beam, suitable for the inspection of larger areas, yielding information on more extended regions. Therefore, TwinMic allowed to evaluate Mg distribution at the single intra-hepatocellular level, while XRF beamline at the intracellular level of a few hepatocytes, complementing each other and providing information at two length scales. Also, the excitation energies were slightly different (1.5 keV for TwinMic, 2.2 keV for XRF). However, the two energies yield similar quantification accuracies. We clarified this in the method section (lines 190-204 of the revised version).
- In Page 5: Determination of Magnesium in hepatocytes by Synchrotron Based X-ray Fluorescence Microscopy: There were no documentations how to select the region of interest and to the way of statistical determination of counts.
The regions to be mapped were evaluated by inspecting the histological sections obtained by staining adjacent slices of tissue with eosin and hematoxylin. XRF spectra were processed by PyMCA software. As far as the statistical validity of the peaks counts, elemental distribution was retrieved by deconvolving and fitting the XRF peaks through a least-squares fitting algorithm. PyMCA software has a built-in feature for reliable background removal. In particular, we used SNIP background subtraction. This algorithm, applied before the peak intensity estimation, ensures that the noise is removed and that the remaining peaks are statistically relevant. More details can be found in Ref # 60 of the revised version. We clarified this in the method section (lines 225-231 of the revised version).
- In Page 5, Methods in Immunohistochemistry analysis for TRPM7 in hepatocytes: Even human salivary gland cells were used as positive control, how authors define intense/average/weak expression in hepatocytes? Moreover, as in Figure 4, why did authors regard total staining as intense or weak, not as intense and average?
We thank this reviewer because in the old version of the manuscript the methods of immunohistochemical analysis for TRPM7 were not clearly written. Immunohistochemical analysis for TRPM7 liver tissue expression was performed using paraffin-embedded sections obtained in both study group B cirrhotic liver specimens and organ donors. Normal parotid gland tissue from a single sample was used as a positive control to ensure quality and consistency of staining and to be confident that hepatocytes from study samples without stained granules in the cytoplasm were indeed considered negative. Due to the diversity between salivary gland ducts and liver parenchyma, we decided not to use comparison with the intensity of staining for TRPM7 in salivary ducts for semiquantitative assessment of TRPM7 expression in hepatocytes. Semiquantitative analysis of hepatocellular TRPM7 was performed after identifying at low magnification (10x), as a first step, the areas of maximum expression, minimum expression and intermediate expression of each sample. In each of these three areas, analysed separately at higher magnification (40x), the percentage of hepatocytes showing a "dot like" stain in the cytoplasm was recorded and we graded the staining as intense or weak. We defined intense staining as a coarse granular cytoplasmic dark brown stain and weak stainining as a finely granular cytoplasmic light brown stain. Hepatocytes without stained granules in the cytoplasm were considered negative. The total number of stained hepatocytes was considered as the sum of the intensely stained and weakly stained ones. Finally, for each sample, the average among the three areas was calculated for the percentage of hepatocytes with intense, weak and total staining. Nuclear and membrane staining, whenever present, was always barely perceptible thus it was disregarded. We clarified this in the method section (lines 233-267 of the revised version) and in the result section (lines 327-333 of the revised version).
- In Page 9, lane 3-6, regarding TRPM7 staining: Did 81% mean 81% of patients? Did up to 31% of hepatocytes indicate that 31% out of total hepatocytes in a certain patient were TRPM7 intense positive? Again, such vague expression or shortcomings are likely due to the fact that Methods in Immunohistochemistry analysis for TRPM7 in hepatocytes in Page 5 did not define parameters of immunostaining rigorously.
We also thank this reviewer because this part was not clear in the old version of the manuscript. In 25 of 31 (81%) cirrhotic liver biopsies from study population B there was intense staining in some hepatocytes, with the maximum percentage of hepatocytes affected by heavy staining of 31% in one patient. Weak staining affected all cirrhotic liver samples and 45% of these had at least half of the hepatocytes affected. In contrast, only 3 of 10 (30%) of the donor control liver biopsies showed intense staining in any hepatocytes, with the maximum percentage of hepatocytes affected by intense staining of 3% in one donor. Weak staining affected all donor liver samples, but only 10% of these had at least half of the hepatocytes affected. We have rewritten these results more clearly (lines 341-348 of the revised version) in light also of what is better explained in the methods (see our response to point 4 raised by this reviewer)
- In Page 10, lane 3-5, Who and how many pathologists diagnosed histology of donor livers (histological sign of inflammation)?
All immunohistochemistry readings for TRPM7 and regarding the presence or absence of mild inflammatory infiltrate in liver donor biopsies were performed by one author (F.L.G.). We have now added this to the method section (lines 267-269 of the revised version) and we better described the inflammation characteristics in the result section (lines 380-381 of the revised version).
- In Discussion, Page 18, lane 535-536: Were Mg content in hepatocytes negatively associated with TRPM7 staining in corresponding tissue sections?
As already stated in the old result section (lines 459-460 of the revised version), in our study we did not find any correlation between TRPM7 expression and Mg content in hepatocytes. We hypothesize that this could be explained by the fact that there are many Mg transporters and, in the individual patient, the contribution of each of them to try to restore cellular Mg content may be different [see Reference 15 of the manuscript]. We have now added this to the discussion (lines 619-623 of the revised version)
Comments on the Quality of English Language
Minor editing of English language required, but no significant issues detected.

Reviewer 2 Report
In general, the authors' study found that the median content of Mg measured in liver biopsies using
atomic absorption spectroscopy at the time of liver transplantation, is about 1/3 lower in cirrhotic patients than in livers of healthy donor. The authors also found that, in cirrhotic patients, the median of the total percentage of hepatocytes expressing TRPM7 in the cytoplasm is 2.6 times higher than in healthy donors. However, there are many problems such as the format of the paper, which need to be carefully checked and corrected. In addition, the quality of the article would be improved if more study on the potential contact between Mg depletion and overexpression of TRPM7.
Author Response
In general, the authors' study found that the median content of Mg measured in liver biopsies using atomic absorption spectroscopy at the time of liver transplantation, is about 1/3 lower in cirrhotic patients than in livers of healthy donor. The authors also found that, in cirrhotic patients, the median of the total percentage of hepatocytes expressing TRPM7 in the cytoplasm is 2.6 times higher than in healthy donors. However, there are many problems such as the format of the paper, which need to be carefully checked and corrected. In addition, the quality of the article would be improved if more study on the potential contact between Mg depletion and overexpression of TRPM7.
We thank this reviewer for his comments. As regards the format of the manuscript, we agree that in the old version the reading was not very fluent and clear.
We therefore better explained, as also requested by reviewer 1, the methods and results related to the measurement of magnesium inside hepatocytes and TRPM7 expression (lines 190-204, 225-231, 233-267, 327-333 and 341-348 of the revised version).
We have also removed table 2 of the old version and moved it to the supplementary material as Supplementary table S1. Of course, in the text of the results we have included the information necessary for a clear reading and contained in that table (lines 428-432 of the revised version).
Finally, we rewrote the part concerning the correlations that remained significant after multivariate analysis in a clearer way in the discussion (lines 569-582 of the revised version).
Regarding this reviewer's comment "the quality of the article would be improved if more study on the potential contact between Mg depletion and overexpression of TRPM7", we fully agree but, unfortunately, we cannot provide direct evidence that the cause of TRPM7 overexpression in the hepatocytes of cirrhotic patients is intracellular Mg depletion. This will be the subject of future studies, also taking into account that intracellular magnesium levels are maintained by other transporters besides TRPM7.

Round 2
Reviewer 2 Report
This reviewer accepts the revised version.